# Ovarian Cancer in Women with Intellectual Disability: Current Data

**DOI:** 10.3390/cancers17050805

**Published:** 2025-02-26

**Authors:** Brigitte Trétarre, Daniel Satgé

**Affiliations:** 1Registre des Tumeurs de l’Hérault, 298 Rue des Apothicaires, 34090 Montpellier, France; 2Oncodéfi, Parc Euromédecine, 209 Avenue des Apothicaires, 34090 Montpellier, France; daniel.satge@oncodefi.org; 3Centre d’Epidémiologie et de Recherche en Santé des Populations INSERM U1295, Toulouse III University, 31000 Toulouse, France; 4UMR 1302 Institut Desbrest d’Epidémiologie et de Santé Publique INSERM, Université de Montpellier, 340093 Montpellier, France

**Keywords:** cancer symptoms, diagnosis delay, intellectual disability, ovarian cancer, treatment difficulties

## Abstract

This article reviews knowledge of ovarian cancer in women with intellectual disabilities. It also evaluates the frequency of this cancer in the department of Hérault in the south of France, which has a cancer registration. Surveys carried out in the general population and in institutions housing women with intellectual disabilities as well as the Hérault Cancer Registry suggest that the frequency of ovarian cancers is similar in women with intellectual disabilities to that in the general population. The survey in Hérault, close to reality because based on a well-defined population, does not find ovarian cancers before the age of 20 and only cancers in adult women with intellectual disabilities. In this collection, ovarian cancers in adult women with intellectual disabilities occurred 13 years earlier (55 years instead of 63 years) than in women without intellectual disabilities. They were discovered at a more advanced stage, which reduces the chances of recovery. It is necessary to keep in mind the risk of earlier ovarian cancer during the medical follow-up of women who have an intellectual disability.

## 1. Introduction

Ovarian cancer (OC) is the eighth most frequent cancer in women and accounts for 4.7% of all female cancer deaths [1]. In the general population, OC occurs mainly in women aged > 50 years and has a poor global outcome, with nearly 46% survival 5 years after diagnosis. This poor outcome, which makes it the most serious gynecological cancer, is due to its frequently late diagnosis, as outcome is highly related to disease stage [2,3]. Current efforts are directed at the early discovery of these tumors, which appears to be the best way to improve the prognosis.

Intellectual disability (ID) is a condition characterized by cognitive impairment (IQ < 70) and limited adaptive behavior emerging in childhood [4]. Persons with ID account for 1 to 2% of the population. Cancers are at least as frequent in persons with ID as they are in the general population [5,6,7]. OC has not been studied thoroughly in women with ID, and there is a need to improve the research and care for OC in these women [8].

For women in the general population, the strongest and undebatable risk factor for OC is a genetic predisposition, as observed in cases with a family history of breast cancer and OC [9]. Mutations in *BRCA1* and *BRCA2*, as well as Lynch syndrome, are responsible for more than 20% of OCs. Identified non-genetic factors are mainly greater height, overweight and obesity, hormonal replacement therapy, and a personal history of endometriosis [10]. In contrast, multiparity, breastfeeding, progestin contraceptive pills, and tubal ligation are protective factors. This genetic predisposition requires an adapted response that may include preventive salpingo-ovariectomy in some women [11]. Overweight and obesity, which increase OC risk, are more prevalent among individuals with ID than in the general population; it is observed in 69% of adults with ID, more in women than in men, and more in persons with mild and moderate ID than in persons with severe ID [12]. A recent meta-analysis indicated that overweight increases the risk of developing OC by 6% and obesity increases the risk by 19% when these weight characteristics are seen during the premenopausal period [13]. Two reproductive features associated with a decreased risk of OC are less protective in women with ID. The first is parity, as women with ID are less frequently mothers and less likely to breastfeed [14]. However, the contraceptive pills taken by women with ID are protective. Applying specifically to women with ID, the use of psychotropic medications, which are more frequently administered to this population, has been suspected to increase the risk of epithelial OC [15]. However, more recent reports have not found an increased risk of cancer linked to antidepressant use [16]. Globally, the co-occurrence of overweight, reduced childbearing, and reduced breastfeeding suggests that women with ID are theoretically at greater risk of OC than women in the general population.

OC is grouped into several different histological types, each associated with specific characteristics in terms of etiology, incidence, survival, risk factors, and age at diagnosis. OC can be classified into three main types: epithelial cancers (i.e., carcinoma), which comprises 90% of all OCs; sex cord–stromal tumors (3 to 6% of cases); and germ cell tumors (0.5 to 5%) [1].

OC usually presents with non-specific symptoms linked to local tumor development, such as pelvic or abdominal pain, urinary symptoms, digestive symptoms, a feeling of fullness, loss of appetite, and weight loss [17,18]. Half of women aged 45 to 70 years consult their general practitioner each year with these symptoms. The diagnosis of OC is difficult [17] and even more complicated in women with ID, because they do not communicate easily and may express pain in an unusual way, often through behavioral changes, such as becoming unusually quiet or unexpectedly hyperactive. This often leads medical professionals to dismiss their behaviors as symptoms of ID [19]. Caregivers may also ignore symptoms, interpreting them as attention-seeking behavior [19].

This article includes the first literature review of OC in women with ID and reports the largest series of OC in this population. We evaluated whether OC is different in women with ID than in women in the general population in terms of frequency, clinical presentation, histological type, age at discovery, or prognosis.

## 2. Materials and Methods

### 2.1. Literature Search

We performed a literature search in PubMed, Google Scholar, Web of Science, and Cochrane Library without limitations on date and language. We used the following key words: “intellectual disability”, “learning disability”, and “mental retardation”. We also searched using particular diseases: “Down syndrome”, “Turner syndrome”, “Tuberous sclerosis”, “ataxia telangiectasia”, “Sotos syndromes”, “William syndrome”, “Rubinstein Taybi syndrome”, “Prader–Willi syndrome”, and “X trisomy”, with “ovarian carcinoma”, “ovarian cancer”, “ovarian adenocarcinoma”, “ovarian malignancy”, “ovarian sarcoma”, “ovarian dysgerminoma”, “ovarian germ cell tumors”, “ovarian teratoma”, “ovarian teratocarcinoma”, “ovarian endodermal sinus tumor”, “ovarian yolk sac tumor”, “ovarian gonadoblastoma”, “ovarian granulosa cell tumor”, “ovarian sex-cord tumor”, “ovarian stromal tumor”, and “ovarian borderline tumor”. Very little is known on the subject, and in the absence of a previous review, we included case reports and abstracts, even when they were poorly documented. We also included data from epidemiological studies on cancer incidence and cancer mortality, as well as institutional experiences with cancer. The inclusion criteria were confirmed ID and confirmed OC, even if poorly documented, in children and adults. There were no limitations on the publication date or language. The exclusion criteria were articles on biological research, therapy-centered protocols, ovarian benign tumors, or cysts.

We identified 1388 abstracts. After eliminating duplicate publications and articles unrelated to the review, we found 162 publications, which allowed us to select 72 articles that met the inclusion criteria (Figure 1).

### 2.2. Experience in Hérault

We also evaluated data from a study conducted to investigate cancer distribution among and the characteristics of persons with ID in the Hérault department of southern France. We looked for OC in women with ID in a list of cancers collected over 15 years (2008–2023). These cases were compared to OC in women in the general population during the same period by extracting information from the local register, the Registre des tumeurs de l’Hérault (Hérault Tumor Registry [HTR]), which covers 1,100,000 persons. The CHAID study (Cancer in Hérault of Adults with Intellectual Disability) was approved by the Commission Informatique et liberté (#913052). According to the rules, patients alive at the time of the study were asked to provide informed consent.

The qualitative variables are presented as the number and percentage and quantitative variables as the mean and standard deviation. The alpha risk was set to 5%. Fisher’s exact test was used for small, expected frequencies, and the Student’s *t*-test was used to compare quantitative variables. Statistical analyses were performed in R software (version 3.6.3).

## 3. Results

### 3.1. Frequency

Seven studies provided information on the association between OC and ID. In Finland during the period 1967–1997, seven patients with OC were found, whereas 5.7 were expected, leading to a standardized incidence ratio (SIR) of 1.2 (95% confidence interval [CI] 0.5–2.5) [5]. In western Australia during the period 1982–2001, three patients with OC were observed, whereas 2.87 were expected, leading to an SIR of 1.04 (95% CI 0.22–3.05) [6]. In the USA, a comparison of the prevalence of reproductive cancer in women with ID and women in the general population in the year 2010 indicated no differences between the two groups [20]. In Sweden, a population-based cohort study conducted between 1974 and 2016 indicated a more than doubled risk of developing OC, but it was not significant, with a hazard ratio (HR) of 2.2 (95% CI 0.8–6.0) [7]. In the UK, the cross-linking of people with ID and the Scottish living in the Scotland Cancer Registry for the years 2011–2019 indicated a significantly increased incidence of OC (SIR = 1.59, 95% CI 1.05–2.42) [21]. A study in Taiwan on cancer risk in children and adolescents with autistic disorders indicated a significant increase in OC (SIR = 9.21, 95% CI 1.12–33.29)***,*** but it was based on only two cases [22]. In Germany, a cross-sectional study using nationwide health insurance data for the year 2019 indicated a slightly increased prevalence of OC, with an odds ratio of 1.26 (95% CI 1.13–1.4) [23]. These reports suggest that the incidence of OC does not differ in women with ID compared with the general population.

Mortality studies have also suggested that OC is not less frequent in women with ID than in non-disabled women. A British study conducted in the Leicester area during the years 1993–2006 found 17 deaths from malignancies in women with ID, 2 of which were from OC [24]. In the Stoke Park Group of Hospitals, the records of deceased patients over two decades (1936–1955) indicated that 2 of the 51 deaths from cancer in females were due to OC [25]. In an autopsy series from 8 institutions for people with ID deceased in California during the years 1979–1986, 3 cases of OC were found among 67 malignancies (4%), and OC ranked second among gynecological neoplasms [26]. Two of the seventy-four neoplasms responsible for deaths among persons with ID recorded in Israel during the years 1991–2005 were from OC [27]. A 4-year (2010–2014) study in the UK indicated an excess (value not indicated) of OC, but the difference was not significant [28]. The Scottish study cited above [21] indicated significantly increased mortality from OC, with an SMR of 2.86 (95% CI 1.66–4.92).

The clinical experience of two institutions provides more detailed information. Among 300 women with ID in a clinic in Michigan, 33 had a surgical procedure, including 2 for OC: a mucinous cystadenoma of borderline malignant potential was treated via hysterectomy and bilateral salpingo-oophorectomy, and a mature teratoma containing an area of squamous cell carcinoma with metastasis was treated via bilateral salpingo-oophorectomy and omentectomy [29]. At the *Rinnekoti* Institution in Finland during the years 1984–1987, the only invasive genital neoplasm among 255 women with ID was 1 case of OC with hepatic metastases [30].

### 3.2. Histological Types

Ovarian tumors measured from 0.5 cm to 23 cm on their longest axis. Among 40 malignancies with indicated histology in the ovaries, the most frequently reported (77%) in children, adolescents, and young women with ID were malignant germ cell tumors. These cases included 25 dysgerminoma, 1 associated with embryonal carcinoma; 4 yolk sac tumors alone or associated with a teratoma; and 1 teratoma that spawned a spindle cell carcinoma. A dysgerminoma developed on a gonadoblastoma in 5 patients [31,32]. Three malignancies (8%) were sex cord–stromal tumors [33]. Ovarian carcinoma was found in only 4 patients (10%). There was also one Burkitt lymphoma [34] and one case of metastatic cervical cancer in the ovary [35]. We did not find particular histological features that could have been different from women in the general population.

### 3.3. Clinical Aspects

The literature review found 41 OCs in women, with a mean age at diagnosis of 17.8 years, ranging from 3 years to 61 years [36]; only 9 were diagnosed after the age of 20 years. The mean age for dysgerminoma was 15.1 years, and for carcinoma 23.2 years.

OC has been reported mainly in case reports of girls and women with various levels of ID, from patients with borderline ID (IQ 70–85) to patients with severe and deep ID (IQ < 35). OC has also been observed in the context of various genetic conditions (Table 1). Down syndrome is the most frequently reported condition associated with dysgerminoma. In addition, germ cell tumors and metastatic cervical carcinoma in the ovary have been described in ataxia telangiectasia. In other genetic conditions, few cases have been reported, mainly dysgerminoma. Although Turner syndrome is associated with the benign tumor gonadoblastoma, which favors the onset of dysgerminoma [37], we could not find a report of a woman with pure Turner syndrome, ID, and OC. Others were reported without a recognized genetic condition, such as cerebral palsy [38] or profound and multiple disabilities [39].

Among the 41 cases of OC, 21 had documented circumstances of diagnosis, 6 were discovered during investigation for another acute or chronic disease and were an incidental finding during surgery for the evaluation of a genetic condition [69] or menstrual cycle disorders. The most frequently cited symptoms were pain (*n* = 11); menorrhagia and bleeding (*n* = 5); digestive symptoms, such as vomiting, diarrhea, and constipation *(n* = 4); urinary symptoms (*n* = 2); tiredness (*n* = 2); increased abdomen size (*n* = 3); palpable abdominal mass (*n* = 3); weight loss (*n* = 2); and unexplained fever (*n* = 1).

### 3.4. Treatment

Treatment varied greatly among the 27 primary OCs for which the information was available. Two were very small tumors measuring < 1 cm, which did not warrant complementary therapy after diagnostic biopsy [69]. Two patients were in an impaired general state that precluded any therapy for their cancer. In three patients, surgery was only performed according to the standard protocol and the usual treatment decisions during the period in which they were seen [31]. Three patients received complementary radiotherapy after surgery [44]. Five patients with dysgerminoma received additional chemotherapy. One patient treated with bleomycine developed respiratory failure. Another patient with germinoma and embryonal carcinoma also received chemotherapy. One patient did not receive complementary chemotherapy due to advanced pulmonary fibrosis. In one patient, bleomycin treatment was discontinued to avoid pulmonary complications. For other cancers (n = 1 lymphoma, n = 2 carcinoma, and n = 4 non-germinoma malignant germ cell tumors), treatment was carried out according to the usual protocols of the period, but often at reduced doses [34]. As the indicated follow-up for treated women is usually very short, it is difficult to evaluate whether treatment modifications worsened outcomes.

### 3.5. Data from Herault

Table 2 lists 6 OCs and 1 ovarian borderline tumor collected in women with ID from among 284 neoplasms in 259 adults (139 men, 120 women; CHAID study). These OCs were diagnosed in women aged 45–65 years (mean age 53.5 years), 13 years earlier than women in the general population (66.8 years). During the period 2008–2023, 1043 cases of OC were diagnosed in the general population of Hérault (2% of all cancers in women), a similar frequency to women in the general population (5%, *p* = 0.1314). The HTR did not identify OC in children with ID.

Table 3 compares data from the literature to observations in the Hérault Tumor Registry.

Cancers were revealed from urinary symptoms and abdominal pain in two patients, digestive symptoms in one patient, respiratory symptoms in one patient, and by a decline in their general condition in two patients. Symptoms were not indicated for two of the seven women. Three patients were at a metastatic stage when their cancer was discovered. Four patients were surgically treated; only one of them received complementary chemotherapy. The patient with a borderline ovarian tumor is still alive. The six patients with OC died. Five deaths were related to the cancer, four occurred between 30 and 49 days after diagnosis, and one at 1 year 4 months. The patient with profound and multiple disabilities died from another cause.

In the HTR, 90% of patients with OC underwent surgical treatment, 79% were treated with complementary chemotherapy, and 1.5% were treated with complementary radiotherapy. In addition, 78% of women treated for OC were alive after 1 year and 45% after 5 years.

## 4. Discussion

The six OCs observed among 120 women with cancer in the HTR suggest a frequency in women with ID of 5%. As such, OC deserves attention and, during medical follow-up, it should be kept in mind that women with ID may develop OC.

All of the scattered and heterogeneous data from epidemiological studies, mortality studies, institutional experience, and Hérault support the notion that OC could be at least as frequent in women with ID as in women in the general population.

Germ cell tumors comprised 77% of our literature review vs. 0.5 to 5% in the general population, whereas only 9% of cases were ovarian carcinoma, vs. 90% in the general population. This discrepancy could be the result of publication bias due to the preferential reporting of cases associated with a particular genetic condition, which usually occur earlier. Two conditions were associated with a particular histological type. Down syndrome had 10 cases of ovarian dysgerminoma in adolescents and young women. Ataxia telangiectasia was associated with four germ cell tumors, including three dysgerminoma. In the 41 cases, we did not observe a particular association with race or specific comorbidities. In contrast, OCs found in adults using the CHAID study comprised carcinoma in five patients for whom the histology was known. The ovarian carcinoma we reported in a woman with Down syndrome is notable because all previously reported cases in Down syndrome were dysgerminoma in girls, adolescents, and young women. Our series may provide more actual information on cancer type distribution than case reports, which present rare and exceptional observations. Additional epidemiological data are needed to achieve better knowledge on the gynecological malignancy burden in women with ID.

The symptoms revealing OC in girls and women with ID did not differ from those in the general population. It is important to develop communication skills adapted to these persons [73]. Screening for OC is not recommended in the general population due to the lack of sensitive and specific biomarkers [74,75]. As OC seems to have the same incidence in women with ID, a particular protocol is not warranted. Nonetheless, clinicians following girls, adolescents, and women with Down syndrome should pay attention to abdominal organs and ask for an ultrasound examination at the slightest doubt. As OC diagnosis is delayed more frequently in women in the general population with a low education level [18], late diagnosis is probable in women with intellectual impairment.

The few data available on OC treatment and its results in women with ID in the literature show surgery without complications in children. However, long-term follow-up is lacking. In adults, no data are currently available on the stage at diagnosis and evolution. Our series indicates that diagnosis is made too late to allow for a curative approach. Thus, it is important to keep in mind that women with ID may develop OC.

As a general rule, consent for treatment should be obtained from the patient with ID when they can understand the proposed treatment. Many patients with ID can be involved in the treatment of their illness and wish to be involved in the decision-making process [19,36]. For persons with profound ID, the family is needed [76].

Successful treatment is hardly possible without the help of professional and familial caregivers. Caregivers know how to communicate with the person, how to explain the situation for her, and how to transmit confidence and help throughout the cancer journey, as shown by a series of 50 interviews with professionals in oncology teams, family members, and professional caregivers [77].

Without clear, adapted recommendations for cancer treatment, medical decision-making is difficult and raises ethical dilemmas. In one-third of cases, cancer treatment in patients with ID is modified, either by dose reduction or complete omission of chemotherapy of radiotherapy, or both [76]. Alternative treatment options, such as intraoperative radiotherapy, have been applied successfully for persons with ID with breast cancer when external beam radiation was difficult [78] or in modified protocols for testicular cancer, even in cases of profound and multiple disabilities [79].

The main limitation of this article is that the review provides a limited number of cases, probably because little attention has been paid to this subject. In addition, the review provides many unusual cases. This could lead to the idea that OC is almost observed in children and adolescents with ID. Therefore, we added population-based data from Hérault, which suggests that, in women with ID, OC is observed mainly in aged women, similar to the general population.

## 5. Conclusions

This first review of OC in girls and women with ID suggests that these tumors are not less frequent in this population than in the general population, and that the histological types do not clearly differ from those in women without ID. In the literature and our population-based study, OC was revealed from symptoms similar to those observed in the general population, but they were not easy to identify given the communication difficulties with individuals with ID. In adults, tumors were discovered 13 years later than in the general population, and at an advanced stage, leading to worse outcomes. Additional studies are needed to better determine the incidence, age at diagnosis, histological types, treatment, and evolution of OC in women with ID. It is important to keep in mind that OC must be found as early as possible to allow for the best outcome.

## Figures and Tables

**Figure 1 cancers-17-00805-f001:**
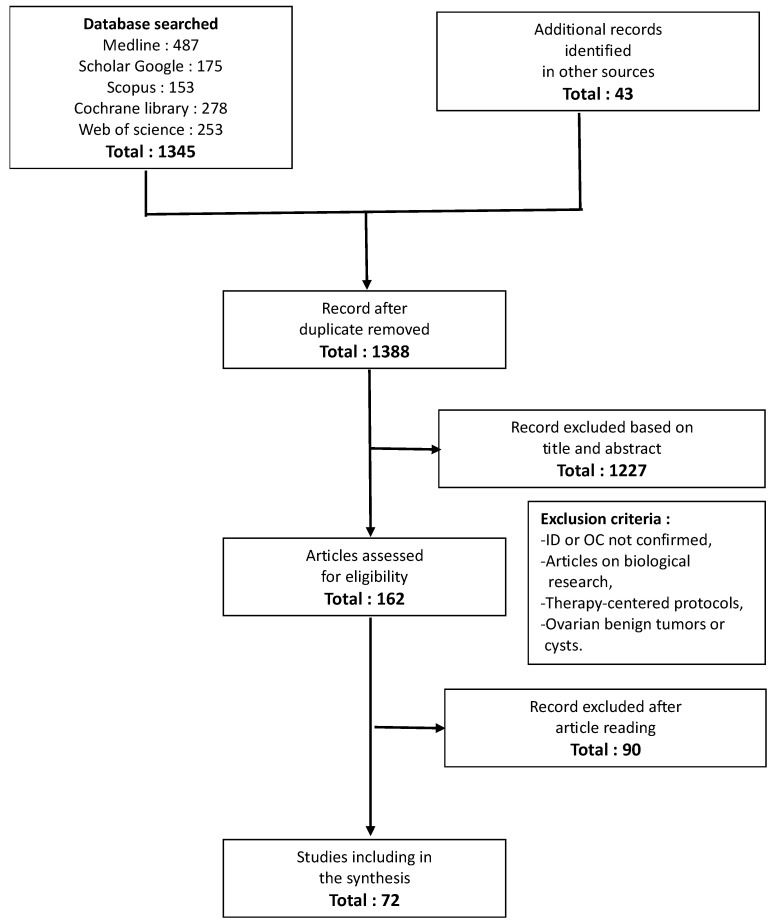
Flow chart of article inclusion.

**Table 1 cancers-17-00805-t001:** Ovarian cancer (OC) in women with intellectual disability (ID) in the context of genetic or cytogenetic conditions.

Genetic Condition	Cancer Type	Age, Years	Reference(s)
Down syndrome	Dysgerminoma (10) Dysgerminoma + EC (1) Ovarian cancer NOS (2)	11–24 14 18 NI	Gesmundo, 1988 [40] Teinturier, 1994 [41] Smucker, 1999 [42] Boker, 2002 [43] Satge, 2006 [44] Girard, 2010 [45] Hasle, 2016 [46] Cepeda, 2022 [47] Samingan, 2024 [48] Ferreira, 2024 [49] Hasle, 2016 [46]
Ataxia telangiectasia	Dysgerminoma (3) YST (1) Metastatic carcinoma (1)	15–34	Dunn, 1964 [50] Goldsmith, 1974 [51] Buyse, 1976 [52] Pecorelli, 1988 [53] Cao, 2020 [35]
Triple X syndrome	Dysgerminoma (2)	22–24	Kemp, 1995 [54] Moskwinska, 2021 [55]
Prader–Willi syndrome	Dysgerminoma developed on sex cord tumor Carcinoma	13 20	Maya-Gonzalez, 2023 [56] Maya-Gonzalez, 2023 [56]
Fragile X syndrome	Gynandroblastoma	17	Lejeune, 2011 [57]
Williams syndrome	Burkitt lymphoma	10	Onimoe, 2011 [34]
Coach syndrome	Cancer	40	Mitsui, 2009 [58]
Nijmegen syndrome	Dysgerminoma	13	Krawczyk, 2021 [31]
Apert syndrome	Dysgerminoma	13	Rouzier, 2008 [59]
Rubinstein Taybi syndrome	Mucinous carcinoma	29	Johannesen, 2015 [60]
Smith–Lemli–Opitz syndrome	Dysgerminoma + YST + EC	19	Patsner, 1989 [61]
Tuberous sclerosis	Juvenile granulosa cell tumor	8	Guo, 2006 [62]
Coffin–Siris syndrome	Small cell carcinoma with hypercalcemia	13	Errichiello, 2017 [63]
Noonan syndrome	Dysgerminoma + TRT + YST	12	Hanson, 2014 [64]
Cowden syndrome	Endometrial carcinoma	31	Matsubayashi, 2019 [65]
Rett syndrome	Dysgerminoma	13	Schultewolter, 2024 [66]
Wolff–Hirschhorn syndrome	Dysgerminoma	10	Schultewolter, 2024 [66]
Trisomy 13	Dysgerminoma	11	Kikuchi, 1999 [67]
WAGR syndrome	Dysgerminoma	24	Miura, 2016 [68]
Trisomy 9p	Dysgerminoma	12	De Ravel, 2003 [69]
Monosomy X, t(Y:18)	Dysgerminoma	3	Cassorla, 1981 [70]
Trisomy 14p	Malignant TRT + YST	16	Lee-Jones, 2004 [71]

YST = yolk sac tumor, EC = embryonal carcinoma, TRT = teratoma, NOS = not otherwise specified, and NI = not indicated. “+” indicates an additional component in the tumor.

**Table 2 cancers-17-00805-t002:** Ovarian carcinoma (OC) in women with intellectual disability (ID) in the Hérault Tumor Registry.

Age, Years	ID Cause	Tumor Size and Side	Histology	Tumor Stage	Treatment, Evolution
57	NA	Right: 5.4 cm Left: 3.3 cm	NA	Stage 4	Palliative care, DOD (30 d)
47	Prematurity-associated psychosis	Right: 24 × 20 cm	Borderline mucinous carcinoma	Stage 1 (pT1a)	Right ovariectomy, alive
45	Down syndrome	4 × 2.5 × 2 cm 4 × 3 × 2 cm ^■^	Grade 3 bilateral papillary serous carcinoma	Stage 4 (pT3C)	Bilateral adnexectomy palliative care, DOD (30 d)
59	Prematurity motor impairment Blindness	NA	NA	NA (cT3c)	DOD before surgery (43 d)
45	Profound and multiple disabilities	Left: 12 × 8 × 5 cm	Grade 1 endometrial carcinoma *	Stage 1 (pT1a)	Bilateral adnexectomy and hysterectomy–omentectomy, death unrelated to the disease
65	NA	Right: 63 cm	Grade 3 papillary serous carcinoma	Stage 4	Palliative care, DOD (46 d)
48	NA	Right: 12 cm Left: 16 cm	Right: borderline tumor Left: mucinous cystadenocarcinoma	Stage 2 (pT2a)	Surgery and chemotherapy, DOD (1 y 4 m)

ID = intellectual disability, NA = not available, DOD = died of disease, y = year, m = months, d = days. ^■^ The side was not indicated. * Also had an associated pT1 endometrial carcinoma.

**Table 3 cancers-17-00805-t003:** Comparison of characteristics of ovarian cancer between the general population in France, the literature review of women with ID, and women with ID in the Hérault Tumor Registry.

		General Population	Women with ID Literature Review n = 41	Women with ID Hérault Tumor Registry n = 6
Age at diagnosis	Adult Child	66 ^†^ –	23.2 * 15.1 ^∆^	53.5 ID (vs. 66.8 GPH) No case
Histological types	Carcinoma SCST Germ cell tumor	90% 0.3–6% 0.5–5%	9% 9% 77%	100% (n = 5) No case No case
Survival	1 year 5 years	74% ^†^ 40% ^†^	Insufficient follow-up	0% –

* Carcinoma only; ^∆^ germ cell tumors; ID = intellectual disability; GPH = general population Hérault; SCST = sex cord–stromal tumor; and ^†^ Trétarre et al. [72].

## Data Availability

Data are available upon request to the authors.

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
