# Peer review of "Ovarian Cancer in Women with Intellectual Disability: Current Data"

_cancers, 2025, doi:10.3390/cancers17050805_

Round 1

Reviewer 1 Report

Comments and Suggestions for Authors

Overall Evaluation: This manuscript is average, with poor organization, unclear language, suboptimal tables, and inconsistent data presentation. Major revisions are needed before publication. Revisions have been grouped below:

Abbreviations/Acronyms:

  • Define abbreviations upon first appearance in the abstract, main text, and first figure or table.
  • Maintain consistent font and size throughout (e.g., lines 41–42, 150–154).
  • Remove unnecessary spaces in numbers (e.g., 1 342 → 1342).
  • Ensure data presentation follows a uniform format (e.g., alphabetical vs numerical order).
  • In Table 1, abbreviate malignant ovarian cancer and intellectual disability appropriately.
  • Define NOS and NI in Table 1.
  • Correct "acute of chronic" to "acute or chronic" (line 149).
  • Use ID instead of DI (line 81) or clarify DI.
  • Correct "dead of disease" to "died of disease" (Table 2).
  • Define DS upon first use (line 238).
  • Remove repeated "IN" (line 279).

Title: The title should include the study type, e.g., "Ovarian Cancer in Women with Intellectual Disability: A Descriptive Study."

Abstract:

  • Follow a structured format without headings.
  • Summarize main findings and conclusions.
  • Address the relationship between intellectual disability and ovarian cancer diagnosis.

Introduction:

  • Define Intellectual Disability with criteria (e.g., IQ < 70).
  • State the study's main conclusion.
  • Introduce ovarian cancer classification earlier.

Materials and Methods:

  • Clearly outline inclusion and exclusion criteria.
  • Clarify the flowchart in Figure 1.

Results:

  • Specify if "seven OCs" refers to types or cases (lines 81, 84, 171).
  • Clarify if "double risk" refers to cancer development or missed diagnosis (lines 88–89).
  • Specify whether findings relate to overall or ovarian cancer (lines 92–94).
  • Remove redundant text (e.g., line 111–113, 116).
  • Use "et al." consistently.
  • Improve presentation of Tables 1 and 2.
  • Clarify ambiguity in cancer types and numerical data (lines 134, 36).

Discussion:

  • Move risk/protective factors section to the Introduction.
  • Add a clear limitations section.
  • Relocate symptom descriptions to the Introduction.

Additional Notes:

  • Update outdated references.
  • Address inconsistencies in line formatting (e.g., 46% survival, pluralistic research questions, sentence restructuring).
  • Clarify the time period in "three decades" (line 101).
  • Rephrase unclear lines (e.g., 199, 274, 279).

Substantial revisions are required to ensure clarity, consistency, and professional presentation.

Comments on the Quality of English Language

A native english speaker should review the manuscript 

Author Response

To rewiever 1:

Thank you for your indications which improve the article presentation. We made the suggested revisions.

Abbreviations/Acronyms:

  • Define abbreviations upon first appearance in the abstract, main text, and first figure or table.

Done

  • Maintain consistent font and size throughout (e.g., lines 41–42, 150–154).

Done

  • Remove unnecessary spaces in numbers (e.g., 1 342 → 1342).

Done

  • Ensure data presentation follows a uniform format (e.g., alphabetical vs numerical order).

In table 1, genetic conditions are presented by frequency order, and publication date by chronological order fora given condition

  • In Table 1, abbreviate malignant ovarian cancer and intellectual disability appropriately.

Done

  • Define NOS and NI in Table 1.

Done

  • Correct "acute of chronic" to "acute or chronic" (line 149).

Done

  • Use ID instead of DI (line 81) or clarify DI.

Done

  • Correct "dead of disease" to "died of disease" (Table 2).

Done

  • Define DS upon first use (line 238).

Done

  • Remove repeated "IN" (line 279).

Done

Title: The title should include the study type, e.g., "Ovarian Cancer in Women with Intellectual Disability: A Descriptive Study."

The new title is: “Ovarian cancer in women with intellectual disability: Current data and a descriptive study”.  We maintained the indication “cancer data” which means a review. It is important since, as far as we are aware, no preceding review has been published.

Abstract:

  • Follow a structured format without headings.

As the journal ask a structured abstract with headings, we did not modify.

  • Summarize main findings and conclusions.

Done

  • Address the relationship between intellectual disability and ovarian cancer diagnosis.

Done

Introduction:

  • Define Intellectual Disability with criteria (e.g., IQ < 70).

Done

  • State the study's main conclusion.

Done

  • Introduce ovarian cancer classification earlier.

Done

Materials and Methods:

  • Clearly outline inclusion and exclusion criteria.

Done

  • Clarify the flowchart in Figure 1.

We indicated reasons for excluded articles. Numbers of the flow chart are modified since we included four additional articles. Two of an OC in women with Down syndrome the third on an adult woman with ID. The fourth is a very recently published epidemiological study on prevalence of OC in women with ID.

Results:

  • Specify if "seven OCs" refers to types or cases (lines 81, 84, 171).

Done

  • Clarify if "double risk" refers to cancer development or missed diagnosis (lines 88–89).

Done

  • Specify whether findings relate to overall or ovarian cancer (lines 92–94).

Done

  • Remove redundant text (e.g., line 111–113, 116).

For line 116: Done

for lines 111-113:  we did not modify the sentences since the two cases are not exactly identical.

.  Use "et al." consistently.

Done

  • Improve presentation of Tables 1 and 2.

Done

  • Clarify ambiguity in cancer types and numerical data (lines 134, 36).

Done

Discussion:

  • Move risk/protective factors section to the Introduction.

Done

  • Add a clear limitations section.

The main limitation of this article is that the review provides a limited number of cases, probably since little attention has been paid to this subject. Also, the review gives many unusual cases. This could lead to the idea that OC in almost observed in children and adolescents with ID. It is why we added population-based data from Hérault which suggests that in women with ID, similarly as in the general population, OC is observed mainly in adult and aged women.

  • Relocate symptom descriptions to the Introduction.

Done

Additional Notes:

  • Update outdated references.

1) For OC cancer screening we added two articles Sideris et al 2022 and Liberto et al 2022

2) For epidemiological data we added the reference Webb et al 2024

3) For OC cancer diagnosis we added the reference Hong et al 2024

  • Address inconsistencies in line formatting (e.g., 46% survival, pluralistic research questions, sentence restructuring).

Done

  • Clarify the time period in "three decades" (line 101).

Done (two decades)

  • Rephrase unclear lines (e.g., 199, 274, 279).

Done

Substantial revisions are required to ensure clarity, consistency, and professional presentation.

Comments on the Quality of English Language

A native english speaker should review the manuscript 

The article has been edited for English.

Reviewer 2 Report

Comments and Suggestions for Authors

This article explores the existing data on ovarian cancer in women with intellectual disability, a topic of significant clinical importance. This population often faces substantial challenges in early screening, diagnosis, and treatment of cancer. The article provides a comprehensive review of the specificities related to the occurrence, diagnosis, and treatment of ovarian cancer in women with intellectual disabilities, synthesizing current research and offering future directions for investigation.

Overall, the manuscript is well-organized and presents the data in a clear and systematic manner. The retrospective analysis of existing data and the policy recommendations are commendable. However, several areas need refinement to improve the article's clarity and scientific rigor.

  1. Outdated Literature Review:
    • The literature review section is comprehensive but lacks references to more recent studies. Although past research is well summarized, the article could benefit from the inclusion of studies published after 2020 to ensure the content remains up-to-date.
    • I recommend adding more recent research to reflect the latest developments in the field, especially studies that focus on diagnostic advancements and therapeutic interventions in this specific population.
  2. Data Presentation and Analysis:
    • While the article discusses the epidemiological features of ovarian cancer in women with intellectual disabilities, the data analysis and statistical methods are somewhat lacking in depth. The manuscript would benefit from a more detailed examination of how different types of intellectual disabilities correlate with ovarian cancer incidence, considering factors such as age, race, and comorbidities.
    • I recommend presenting the data in more detail using tables, figures, and visual aids to enhance clarity and allow readers to grasp the study outcomes more easily.
  3. Clinical Practice Guidance:
    • Although the article touches on early diagnosis and screening methods for ovarian cancer, it lacks specific clinical recommendations, particularly in the context of women with intellectual disabilities. It would be helpful to include practical guidelines on how to adapt screening tools and diagnostic approaches for this vulnerable population.
    • Consider discussing case studies or examples of clinical scenarios to provide more practical advice for clinicians who may face these challenges in everyday practice.
  4. Ethical and Social Considerations:
    • The manuscript does not delve into the ethical and social issues surrounding the diagnosis and treatment of ovarian cancer in women with intellectual disabilities. Issues such as informed consent, medical decision-making, and the role of caregivers or family members are vital in this context and should be addressed more thoroughly.
    • I suggest expanding the discussion to include the ethical dilemmas clinicians may face when treating this group of patients and the broader societal implications.

In conclusion, this article provides an essential perspective on ovarian cancer in women with intellectual disabilities and adds valuable insight to the existing literature. By addressing the points mentioned above—such as updating the literature review, enhancing data analysis, providing clinical recommendations, discussing ethical issues, and refining the language—the manuscript could become a more comprehensive and impactful contribution to the field. I recommend that the authors revise the manuscript according to these suggestions and resubmit for further review.

Author Response

To reviewer 2

We thank you for the constructive comments which improve the manuscript and provide important and useful practical indications.

  1. Outdated Literature Review:
    • The literature review section is comprehensive but lacks references to more recent studies. Although past research is well summarized, the article could benefit from the inclusion of studies published after 2020 to ensure the content remains up-to-date.

As far as we are aware the review is complete for epidemiological data. Many epidemiologic works on cancer in ID do not provide indications on OC . Nonetheless, in the mean time we could find an abstract on cancer prevalence in persons with ID very recently reported from Germany [Sappok et al 2024]. We also added two articles reporting a dysgerminoma in adolescents with Down syndrome [Samingan, 2024, Ferreira et al 2024], and one cancer in an aged woman with ID [Flynn et al 2016].

    • I recommend adding more recent research to reflect the latest developments in the field, especially studies that focus on diagnostic advancements and therapeutic interventions in this specific population.

We updated the literature adding recent articles on OC screening in the general population [Liberto et al 2022, Sideris et al 2024]. No article focused on OC screening is available for women with ID. Similarly, no particular work has been dedicated to OC diagnosis in women with ID.

For treatment, we added articles which present treatment difficulties, and the need to adapt usual protocols [Boonman et al 2022, Bhimani et al 2024, Delany et al 2023]. None of these publications have focused on OC. When treating persons with ID, particularly the need to adapt protocols for breast cancer and testicular cancer.

While the article discusses the epidemiological features of ovarian cancer in women with intellectual disabilities, the data analysis and statistical methods are somewhat lacking in depth. The manuscript would benefit from a more detailed examination of how different types of intellectual disabilities correlate with ovarian cancer incidence, considering factors such as age, race, and comorbidities.

We added a statistic evaluation of OC frequency for the period 2008-2023 compared to the general population in Hérault indicating a similar frequency (5% p=0.1314). A comparison of age at discovery between OC in women with ID and women of the general population shows a 13 years younger age (53.5 vs 66.8 5% p=0.0116).

The sentence has been re-written: ‘’two conditions were associated with a particular histological type’’. Down syndrome (DS) with 10 ovarian dysgerminoma in adolescents and young women, whereas carcinoma which appears later at the fith and sixth decades. Ataxia telangiectasia was associated with four germ cell tumors including three dysgerminoma. We did not observe in the 41 cases a particular association with race nor which special comorbidities.

    • I recommend presenting the data in more detail using tables, figures, and visual aids to enhance clarity and allow readers to grasp the study outcomes more easily.

A table

  1. Clinical Practice Guidance:
    • Although the article touches on early diagnosis and screening methods for ovarian cancer, it lacks specific clinical recommendations, particularly in the context of women with intellectual disabilities. It would be helpful to include practical guidelines on how to adapt screening tools and diagnostic approaches for this vulnerable population.

We have added

Screening for OC is not recommended in the general population due to the lack of sensitive and specific biomarkers [Liberto et al 2022, Sideris et al 2024]. As in women with ID OC seem to have the same incidence, a particular protocol is not warrented. Nonetheless clinicians following girls, adolescents and women with DS should pay attention to abdominal organs and ask an ultrasound examination at the slightest doubt.

    • Consider discussing case studies or examples of clinical scenarios to provide more practical advice for clinicians who may face these challenges in everyday practice.

We agree that such examples would be of interest. We have added some general indications, see below.

  1. Ethical and Social Considerations:
    • The manuscript does not delve into the ethical and social issues surrounding the diagnosis and treatment of ovarian cancer in women with intellectual disabilities. Issues such as informed consent, medical decision-making, and the role of caregivers or family members are vital in this context and should be addressed more thoroughly.
    • I suggest expanding the discussion to include the ethical dilemmas clinicians may face when treating this group of patients and the broader societal implications.

These points have been addressed in the following sentences added in the discussion.

As a general rule, treatment consent should be obtained from the patient with ID when she/he can understand the proposed treatment. Illness and treatment can be explained to many patients with ID, and they wish to be involved in the decision-making process [Flynn et al 2016, Tuffrey-Wijne et al 2006]. For persons with profound ID, the family is needed [Boonman et al 2022].

The role of professional and familial caregivers is essential. A successful treatment is hardly possible without this help. Caregivers know how to communicate with the person, how to explain her/he the situation, how to transmit confidence and help them throughout the cancer journey. This was shown in a series of 50 interviews of professionals in oncologic teams, family members and professional caregivers (Sarah Habib-Hadef et al, communication at the International Psycho-Oncology Society congress, Berlin 2017 -article submited)

Without available clear adapted recommendation for cancer treatment, the medical decision is difficult and raise ethical dilemmas. In one third of cases cancer treatment of patients with ID is modified, either by dose reduction, complete omission of chemotherapy of radiotherapy, or both [Boonman et al 2022]. Alternative treatment options, for example intraoperative radiotherapy, has been successfully given to persons with ID having a breast cancer when external beam radiation was difficult [Bhiman et al 2024] or modified protocols for testicular cancer, even in case of profound and multiple disability [Delany et al 2023].

Reviewer 3 Report

Comments and Suggestions for Authors

Ovarian cancer in women with intellectual disabilities: current data reported by Brigitte Trétarre and Daniel Satgé,

report a synthesis of 68 literature reviews concerning the announced objective.

The authors report that ovarian cancer could be as common in women with intellectual disabilities In the ages of 16.8 years, ranging from 3 years to 40 years.

In parallel, as a control in the general population, data from the Hérault Tumor Registry (HTR), in the south of France, were used. This work is sufficiently documented concerning the origins of intellectual disabilities and also the histopathology of cancer, even it is a literature review, is original for an up to date data in these types of patients.

So it is an interesting work. Consequently this report can be accepted in its current state for publication in the journal cancers.

Author Response

To reviewer 3

We thank reviewer 3 for his generous evaluation of the article. We agree that the review of the literature is up to date.